# In situ field measurements of the temporal evolution of low-frequency sea-ice dielectric properties in relation to temperature, salinity, and microstructure

Megan O'Sadnick[1,5], Malcolm Ingham[2], Hajo Eicken[3], Erin Pettit[4]

[1]Geophysical Institute, University of Alaska Fairbanks, Alaska, USA
[2] School of Chemical and Physical Sciences, Victoria University of Wellington, Wellington, New Zealand
[3]International Arctic Research Center, University of Alaska Fairbanks, Alaska, USA
[4]Department of Geosciences, University of Alaska Fairbanks, Alaska, USA
[5]Northern Research Institute (Norut) Narvik, Narvik, Norway

*Correspondence to*: Megan O'Sadnick (mosadnick@gmail.com)

**Keywords.** Sea ice, microstructure, complex dielectric permittivity, low-frequency electric measurements, in situ, seasonal evolution

**Abstract.** The seasonal evolution of sea-ice microstructure controls key ice properties, including those governing ocean-atmosphere heat and gas exchange, remote-sensing signatures and the role of the ice cover as a habitat. Non-destructive in situ monitoring of sea-ice microstructure is of value for sea-ice research and operations, but remains elusive to date. We examine the potential for the electric properties of sea ice, which is highly sensitive to the brine distribution within the ice, to serve as a proxy for microstructure and, hence, other ice transport properties. Throughout spring of 2013 and 2014, we measured complex dielectric permittivity in the range of 10 Hz to 95 kHz in landfast ice off the coast of Barrow, Alaska. Temperature and salinity measurements and ice samples provide data to characterize ice microstructure in relation to these permittivity measurements. The results reveal a significant correlation between complex dielectric permittivity, brine volume fraction, and microstructural characteristics including pore volume and connectivity, derived from x-ray microtomography of core samples. The influence of temperature and salinity variations, as well as the relationships between ice properties, microstructural characteristics, and dielectric behavior emerge from multivariate analysis of the combined data set. Our findings suggest some promise for low-frequency permittivity measurements to track seasonal evolution of a combination of mean pore volume, fractional connectivity, and pore surface area-to-volume ratio, which in turn may serve as proxies for key sea-ice transport properties.

## 1 Introduction

Sea ice covers a significant fraction of the polar oceans for much of the year. Ice extent ranges between 3.4 and 15 million km$^2$ in the Arctic and 2.3 and 20 million km$^2$ in the Antarctic (Fetterer et al., 2016)The ice canopy controls air-sea exchange, such as greatly increasing surface albedo or damping surface waves; it also impacts the presence and movement of sea life.

In contrast to freshwater ice, the microstructure of sea ice is characterized by brine pores and channels which evolve in size, shape, and spatial arrangement from initial ice formation through melt. These features govern the thermal and mechanical properties of the ice and resultantly impact its behaviour on the macro-scale (Petrich and Eicken, in press). In addition, brine pores and channels provide an important habitat for microbiota, which draw on nutrients provided through convection of seawater throughout the lower layers of the ice and serve as an important part of the polar oceans' food webs (Gradinger et al. 2010). The dispersal of pollutants released under the sea ice is also largely controlled by its microstructure, with brine

volume fraction and pore connectivity playing a key role in the upward migration and potential surfacing of pollutants such as oil (Karlsson et al., 2011).

Given the impact of sea ice on its surrounding environment, an understanding of its behavior at the micro- and macro-scale is vital. Observing the temporal evolution of sea ice is challenging however, as any removal of ice cores to obtain data on ice properties and microstructure results in the loss of brine and alterations of pore microstructure. The remoteness of field sites also limits sampling campaigns and decreases the observations made during critical transitions in the ice. Methods to observe ice properties and microstructural evolution in situ and continuously are therefore necessary to provide a continuous, undisturbed record. One promising approach is to use the dielectric properties of sea ice as a proxy for the brine quantity and distribution and pore space connectivity. Specifically, laboratory studies suggest that at low frequencies, defined here as those below 10 kHz − the Debye relaxation of pure ice, dielectric properties may be a more sensitive indicator of ice microstructure than at higher frequencies due to the occurrence of space charge polarization at pore boundaries (Buchanan et al. 2011).

Here we explore the relationship between low-frequency complex dielectric properties of sea ice with ice properties such as temperature, salinity and brine volume fraction and specific aspects of ice microstructure. We present the first in situ measurements of the seasonal variation of low-frequency complex permittivity of natural sea ice, measured at Barrow, Alaska in 2013 and 2014. While Ingham et al. (2012) presented measurements of the complex permittivity of Antarctic sea ice, they did not track its temporal evolution. We also show associated measurements of ice properties including temperature, salinity and brine volume fraction; and microstructural characterization of sea-ice samples gathered in parallel with the impedance measurements. We find the frequency dependence of complex permittivity varies seasonally and seek to understand the physical mechanisms dominating the seasonality of this dependence. Toward this goal, we investigate correlations between the permittivity data and the various ancillary measurements to assess the potential use of low-frequency permittivity measurements as a proxy for sea-ice microstructure.

## 2 Background

Research into the electric properties of sea ice and their link to microstructure was first conducted in the 1970s (Addison, 1969, 1970; Vant et al., 1978; Milton, 1981). While these studies relied on simplistic assumptions about ice microstructure, they evaluated the relevant physical processes contributing to the real ($\varepsilon^{'}$) and imaginary ($\varepsilon^{''}$) parts of the dielectric permittivity. In recent years, potential relevance for satellite remote sensing (Arcone et al., 1986, Hallikainen and Winebrenner, 1992) and in situ monitoring of ice salinity and brine volume fraction (Backstrom and Eicken, 2006, Notz and Worster, 2008, Pringle et al., 2009a), have motivated dielectric measurements at GHz frequencies. This prior work demonstrated that as frequency decreases to less than 100 MHz, dielectric measurements are increasingly sensitive to the distribution, shape, and size of brine pores enclosed in the ice matrix (Morey et al., 1984). However, with the exception of work by Addison (1969, 1970) the relationship between low-frequency measurements and microstructure remains largely unexplored.

Other studies of the dielectric properties of sea ice include a study by Ingham et al. (2008) examining surface resistivity measurements in relationship to microstructural anisotropy in columnar ice. They tie errors in estimates of ice thickness

derived from surface resistivity to ice anisotropy and link the error to significant differences in the horizontal versus vertical resistivity. Building on this finding, Jones et al. (2010) applied a cross-borehole technique to track ice anisotropy and the formation and growth of brine channels more closely; their results show a relationship between resistivity, the temperature of the ice, and time of year. Jones et al.'s (2010) study supports the ability of electrical measurements to track the connectivity of brine pores as the ice nears the percolation threshold (Pringle et al., 2009b). This important transition triggers the draining of surface melt ponds, the initiation of sea-water convection throughout the ice volume, and a change in the thermal regime of the ice. Jones et al. (2012) subsequently created a theoretical structural model of sea ice to link measurements of resistivity to the connectivity of the brine pores. They were successful in modelling temporal changes in the relative size of pores and their connectivity. Even at low temperatures however, in order to match measurements of the formation factor (i.e., the ratio of bulk resistivity to brine resistivity), the model required brine connectivity in both the horizontal and vertical directions - due possibly to conduction along ice crystal boundaries rather than through connected pores.

These previous studies suggest an increase in the sensitivity of electrical measurements to variations in sea-ice microstructure as frequency decreased. This relationship, however, was not studied in detail until Buchanan et al. (2011) examined the low-frequency electric properties of sea ice by measuring the complex permittivity of laboratory grown sea ice over a frequency range of 10 Hz to 1000 kHz. By applying a broadband regression model to measurements, Buchanan et al. (2011) obtained estimates of the frequency-independent conduction and bulk polarization and loss. They went on to speculate that trends in measurements of $\varepsilon'$ observed at the lowest frequencies may be due to build-up of charge at the pore/ice interface. This observed trend and the underlying theory suggest polarization effects associated with charge build up may be used as a proxy parameter for the microstructural evolution of sea ice. Lacking in Buchanan et al. (2011)'s study, however, are actual measurements of microstructural characteristics such as the geometry of pore spaces and its variation with temperature. In addition, laboratory grown ice while appropriate to obtain general limits on the permittivity of sea ice, may differ from natural sea ice, in particular thicker ice. We build on these previous findings to study the permittivity-microstructure relationship in situ.

## 2 Methods

### 2.1 Site location

We collected permittivity measurements on landfast sea ice adjacent to a long-term mass-balance measurement site installed approximately 8 km northeast of the Ukpeagvik Inupiat Corporation-National Arctic Research Laboratory (UIC-NARL) base in Barrow, AK (Figure 1; Druckenmiller et al., 2009). The study location provided for undisturbed, gradual in situ freezing and a homogeneous sea-ice structure with sea-ice crystal lamellae oriented perpendicular to the alongshore current. Little ice deformation occurs in this region, resulting in relatively flat ice topography and consistent snow depth.

### 2.2 Permittivity measurements

We derived values of apparent permittivity from measurements of impedance and phase using the cross-borehole technique and instrumentation described by Ingham et al. (2012). The electrode array was comprised of four 2.0 m long electrode strings installed vertically in the sea ice at the corners of a one meter by one meter square. On each string, electrodes consisted of marine grade stainless steel washers positioned at 0.1 m vertical increments. The strings were placed into boreholes drilled in the sea ice in early January with ice growing downwards and embedding electrodes progressively over

the course of the season. In 2013 and 2014, we visited the site and collected data three times after the January installation to capture the microstructural evolution of the ice as it underwent the transition from the cold ice-growth season into spring and summer melt. The first set of measurements each year was taken in late March, with 0.48 m and 0.51 m of new ice growth, respectively, accreted at the bottom after emplacement of electrodes. The second set of measurements in mid-May we timed to capture early warming and the onset of meltwater percolation at the surface. The third set, taken in early June, was intended to capture the ice state after the percolation threshold had been surpassed throughout the ice thickness, which results in substantial meltwater flushing and desalination.

Impedance and phase were measured by passing current ($I$) between two selected electrodes of the same depth but in different boreholes while simultaneously measuring the potential difference ($\Delta V$) through the ice between electrodes 0.1 m above and below. For example, impedance and phase for a depth in the ice of 0.15 m were obtained by injecting current at 0.10 m depth and measuring the potential difference between electrodes at 0.20 m depth. This combination was then reversed with current being injected at 0.20 m depth and the potential difference measured at 0.10 m. We repeated this measurement for each of the six combinations of pairs of electrode strings allowing the mean complex apparent permittivity and related standard deviation at that depth to be calculated from up to 12 independent measurements. To calculate the complex apparent conductivity ($\sigma^*$) from the measured impedance magnitude (the ratio of the magnitudes of $\Delta V$ and $I$) and the phase difference between the measured potential and current, we multiply a geometric factor related to the electrode positions. The complex relative apparent permittivity ($\varepsilon^*$), then, relates to this apparent conductivity through

$$\varepsilon^* = \frac{\sigma^*}{i\omega\varepsilon_o}$$

(1)

where $\omega$ is the angular frequency of the injected current, $\varepsilon_o$ is the permittivity of free space, and $i = \sqrt{-1}$. We measured this complex relative apparent permittivity at 13 discrete frequencies between from 10 Hz and 95 kHz.

Before further analysis, we edited measurements to remove any physically unreasonable data. Because the current must lead the voltage in phase, a positive phase measurement indicates instrumental uncertainty or human error. Such measurements can occur when the impedance magnitude is low, which increases errors in determining the phase. Measurements clearly affected by electrode polarization were also excluded from analysis. Such polarization effects are less obvious in individual data points and require examination of weather conditions, ice properties, and comparison of complex permittivity measurements between months. If brine is in contact with an electrode an ionic double layer can form around the electrode when a current is applied resulting in a substantial increase in polarization, particularly at the lowest frequencies. In laboratory based studies, measurements of permittivity taken on brine of a given salinity can be used to determine the magnitude of the electrode polarization and assist in correcting measurements. Such an approach cannot be taken in the field where brine salinity is approximated and varies greatly throughout the depth of ice and with the seasons. Electrode polarization was found to be a problem in the upper 0.35 m of ice for measurements made in late May 2014, and for all measurements from June 2014. As a result, we excluded these measurements from further analysis.

## 2.3 Measurements of snow and ice properties

In order to derive a relationship between the permittivity and microstructure, we collected ice cores in parallel with the impedance measurements. These ice cores provide estimates of ice temperature, bulk ice salinity, and brine volume fraction throughout the depth of the ice. Cores were drilled within 2 m of the permittivity array and supplemented by measurements gathered at the University of Alaska Fairbanks (UAF) mass-balance station located 15 m away. Sections of the core 0.05 cm in thickness were bagged in the field and brought back to the UIC-NARL campus. Once melted, measurements of bulk ice salinity were made using a YSI Model 30 © handheld salinity, conductivity, and temperature meter. The impact of air temperature on ice temperature measurements varied between field trips leading to the need to compare measurements to those gathered by a string of thermistors frozen vertically into the ice at UAF mass-balance site. As many core measurements varied from these readings by upwards of 3 °C, we determined the temperature measurements from the mass-balance station to be of better use. Brine volume fraction was calculated with expressions from Cox and Weeks (1983) using measured values of temperature and salinity. Air volume fraction was not calculated as they require ice density measurements, which were not gathered here due to field sampling time constraints.

Error associated with temperature, salinity, and calculated values of brine volume fraction can be attributed to measurement error and spatial variability. The relative error of the conductivity meter is estimated to be no greater than +/- 2 %, with this error being substantially less with instrument .calibration and proper handling. Measurements of temperature gathered from the mass-balance site have an absolute error of +/- 0.1 °C. In addition, variations in snow depth can cause lateral variability in ice temperature and salinity. While it is difficult to estimate, we assume here an error of +/- 0.3 °C in temperature and +/- 0.5 ppt in salinity due to spatial variability. To estimate error in values of brine volume fraction, we use actual measurements to define upper and lower bounds of temperature and salinity combinations. For ice with a temperature of -11 °C and salinity of 6 ppt, the absolute error in measurement of brine volume fraction is +/- 0.4 %. As temperature increases, error tends to increase as well. For ice with a temperature of -2.0 °C and salinity of 4 ppt, the absolute error associated with estimates of brine volume fraction is +/- 2.1 %.

## 2.4 Microstructural analysis

We collected additional ice cores for microstructural analysis including pore volume, pore surface area, and fractional connectivity. Once removed, the cores were examined to identify 0.06 m sections representative of different ice textures and pore microstructures. Typically at least one sample was obtained from the top 0.20 m where granular ice is most likely present, a second at around 0.30-0.40 m depth where granular ice transitions to columnar ice, and a third at greater depth where columnar ice is well developed. All samples were brought promptly back to the laboratory in an insulated container where they were centrifuged to remove all brine from open pores and channels. Each sample was then wrapped with aluminium foil, placed in a sealed polyethylene bag and stored at -40 °C to preserve pore microstructure for further analysis after transfer in Dewar vessels to the UAF Geophysical Institute Sea Ice Laboratory.

To quantify pore-space characteristics including the distribution of pore volumes, the overall geometry of the pore space (defined by surface area to volume ratio), and pore connectivity, we imaged ice samples using X-ray computed tomography (CT) techniques. To do so, we used a bandsaw and lathe cooled to between -10 and -15°C to cut ice samples down to a cylinder 50 mm in length and 30 mm in diameter. A Skyscan 1074 portable micro-CT scanner was used for x-ray

tomographic imaging at -20˚C. We gathered individual vertical projection images at 0.90° increments over ~180° totaling 201 projections for each sample. These 16-bit TIFF images with pixel size of 40 μm x 40 μm were reconstructed using NRecon software (Bruker microCT, Kontich, Belgium). Gray values in the images are based on the attenuation of x-rays through the sample and are referred to as radiodensity. For the samples of sea ice imaged, the spread of radiodensities of ice, brine and air extended from 0 to 1600 however artifacts occasionally appeared with radiodensities greater than this upper value. When converting the projection images to a 3D volumetric stack of 8-bit JPEG images, we set bounds to include only pixels with radiodensities in this range.  To extract quantitative data on the microstructure of each ice sample, we further processed and analysed the tomographic data using a maximum likelihood classification to discriminate between air and ice. Using gaussian curves to fit the distribution of radiodensity values, we set the threshold delineating air from ice for all images at a digital number of 91, the value where the two tails from the distributions intersected.  We used this value for all projection images because a consistent method was used to collect and process images. The mode for the gaussian fit for air has a digital number of 45 and for ice has a digital number of 147, therefore, at least 55%  of a voxel equal to 33 $\mu m^3$, must be filled with air to have a digital number below the ice-air threshold. As contrast between these two phases decreases, the minimum volume of a detectable feature will increase and vice-versa. While pore throats and crystal boundaries may at times be smaller in size than this minimum-detectable volume, brine pores and channels are often orders of magnitude larger. Therefore, for the purpose of this study focused largely on bulk properties and trends in microstructural evolution, we did not further address these sources of error.

Determining the threshold delineating ice and brine presented a greater challenge. Due to variations in salinity, brine displays inconsistent radiodensity resulting in a wide range of gray values. In addition, regions of brine are much smaller than those for air and ice leading to a smaller sample size, introduction of mixed voxels and a less apparent mode. By manually identifying pixels known to be brine in combination with examination of the original histogram, an approximate threshold set at a digital number of 160 was applied across all samples. We found this approximation to be sufficient for identifying the small amount of brine present primarily in samples gathered during the coldest periods when disconnected pores prevent some brine from draining during centrifuging. Once thresholds were determined, the image was binarized by setting all pores of either air or brine to 1 (assuming all air pores had brine while in situ), and ice to 0. Through stacking of 2D binarized images, a 3D image of each ice sample was obtained (Figure 2).

For the purpose of this study, it was imperative to obtain quantitative descriptions of individual pore spaces. We used the MatLab Image Processing Toolbox for analysis. For each stack of images representing one ice sample, we defined pores as clusters of 26 connected voxels (voxels sharing a face, edge, or corner) and measured the individual pore volume to surface area ratio, mean pore volume and surface area, and connectivity amongst pores. For the latter, fractional connectivity describes the overall extent of pore networks. As defined by Pringle et al. (2009b), this characteristic is the percentage of pores present at a given depth connected through any path to the uppermost (surface) layer. For subsequent analysis, the depth at which 25 % fractional connectivity occurred was extracted for correlation to permittivity results. In addition, we determined the brine volume fraction of each sample, which at times differed from measurements of bulk brine volume fraction obtained from larger core samples. To further analyze the relationships between ice properties and microstructural characteristics, we applied basic Matlab functions to determine correlation coefficients and perform a principal component analysis.

## 3 Results and Analysis

### 3.1 Air temperature, snow depth and ice thickness

Figure 3 shows air temperature, snow depth, and ice thickness for both 2013 and 2014 from January, when the mass-balance site and electrode strings were emplaced, through June, when all instruments were removed. Until the first week of April, conditions are similar in both 2013 and 2014 with air temperatures ranging between -10 and -35 °C. During this time ice thickens at a constant rate, with ice growing slightly faster in 2014 compared to 2013. Starting in mid-April, however, air temperatures measured in 2013 and 2014 deviated significantly. Except for a brief period in mid-May, air temperatures in 2013 remain below freezing until May 21. After this time, ice growth stagnates and snow depth decreases. Ice begins to thin on June 2. In 2014, on the other hand, air temperatures show their first abrupt rise to near freezing on April 13. The first persistent above freezing temperatures occur on May 1 accompanied by substantial snowfall and subsequent snow melt while temperatures remain above freezing for 2-3 days. During this warming event ice begins to thin, a full month earlier than in 2013. After May 21, 2014, mass balance instruments stopped taking measurements due to technical problems until final data were collected on their removal on June 8. During this time, ice thickness did not decrease substantially because of a presumed drop in air temperature to below freezing between May 21 and June 8, 2014.

### 3.2 Ice temperature, salinity and brine volume fraction

The ice properties shown in Figure 4 reflect the contrasting weather conditions in 2013 and 2014. In the March 2013 data set, ice temperatures in the upper 1.0 m are nearly 3 °C lower than in the March 2014 data set. Despite differences in ice temperature, bulk salinities are similar between the two years, ranging between 4 and 6 ppt. Brine volume fractions reflect the contrasting ice temperatures with values in the upper 1.0 m of ice between 2 – 4 % in 2013 compared to consistently near 5 % in 2014.

May measurements, in particular, express the strong contrast in weather conditions recorded between 2013 and 2014. In the May 2013 data set, ice temperatures are lowest at the top of the ice and increase linearly to the freezing point of seawater at the ice/ocean interface. In May 2014, on the other hand, little variation in temperature is seen throughout the ice, with the upper 1.0 m being no more than 1 °C less than values measured at the bottom of the ice. Measurements of salinity in May 2013 also differ from those gathered in May 2014 but only in the upper 0.4 m of ice. In 2013 salinity at 0.10 m depth is above 8 ppt while in 2014 salinity at the same depth is lower than 2 ppt. From a depth of 0.4 m onwards, salinity in 2013 and 2014 hovers between 4 - 5 ppt. Values of brine volume fraction in May 2013 are lower than values found in May 2014. Brine volume fraction reaches 8 % at 0.1 m depth in 2013 in comparison to a value of 10 % in 2014. Below 0.3 m depth, brine volume fraction in 2013 drops to values of 5 % while in 2014, values averaged nearer to 7 %.

By June, the ice temperatures in both years' data sets are similar in the upper 0.50 m. At greater depths, however, temperatures remain lower in 2013 than in 2014. Salinities in both years are similar with an increase from just above 0 ppt to 4 ppt in the upper 0.50 m. Derived brine volume fractions are high throughout the upper 0.50 m in 2013, with values decreasing with depth. In June 2014, brine volume fraction increases linearly from a minimum at the very top to a local maximum at 0.45 m, with a further increase below 1.0 m depth.

### 3.3 Ice dielectric properties

Figure 5 shows both the calculated real, $\varepsilon'$, and imaginary, $\varepsilon''$, parts of the complex apparent permittivity, $\varepsilon^*$, as a function of frequency. In both the 2013 and 2014 data, the real part of the complex permittivity, $\varepsilon'$, increases as the season progressed for frequencies below 1000 Hz, a trend similar to that seen with rising temperature by Buchanan et al. (2011). The magnitude of this increase varies with depth. In 2013, the increase is greatest above 0.45 m where ice is most sensitive to variations in atmospheric conditions and ice texture transitions from granular to columnar. As depth increases to 1.05m, the difference between March 2013 to June 2013 for the real part, $\varepsilon'$, decreases. Measurements made of the real part, $\varepsilon'$, in 2014 show substantially higher values in comparison to $\varepsilon'$ in 2013. Similar trends persist, however, with the real part, $\varepsilon'$, increasing in magnitude from March 2014 to May 2014 at frequencies below 1000 Hz. A strong depth dependence is not apparent however. The dielectric relaxation of the ice is marked by an inflexion in the real part $\varepsilon'$ at a frequency of around 10 kHz in both 2013 and 2014.

The imaginary part of the permittivity $\varepsilon''$ is directly related to the ionic conductivity of the ice and a clear increase in $\varepsilon''$ is seen across all frequencies between March and June in 2013 and March and May in 2014. Similarly to values for the real part $\varepsilon'$, values for the imaginary part $\varepsilon''$ in March 2014 are substantially higher than those for the previous year.

### 3.4 Relationships between complex permittivity, ice properties, and microstructural characteristics

Buchanan et al. (2011) fit a broadband mathematical model to derived values of complex conductivity to quantify DC conduction, dielectric relaxation, space charge polarization and related electrical parameters. They show a relationship between these properties and temperature which they attribute to the connectivity of pore space and ice microstructure. Buchanan (2011) subsequently applies the dispersed ellipsoid, conductive-dielectric mixture model of Vant et al. (1978) to further define the possible impact of pore geometry and aspect ratio on electric measurements. These results provide some basis for a physical interpretation of electric measurements but lack a quantitative description of sea ice microstructure.

We applied the mathematical model of Buchanan (2011) to field measurements of the temporal evolution of permittivity as a first step to compare and connect behavior of laboratory grown sea ice to that of natural sea ice. Due to the lower spatial density of measurements and larger spread of values for impedance and phase at temperatures between -5 °C and 0 °C, however, the model often did not converge or yielded inconsistent results. To address this challenge and to enhance our understanding of the interrelationships between ice properties, microstructure, and field measurements of complex dielectric permittivity, we present a correlation analysis. Results are intended to guide future field campaigns and aid understanding of the physical and microstructural parameters and processes controlling electric measurements.

Figure 6 through 8 show the real and imaginary parts of the complex permittivity $\varepsilon'$ and $\varepsilon''$ as a function of the measured temperature and bulk ice salinity, and the calculated brine volume fraction of the ice, at frequencies of 10, 100, 1000 and 10000 Hz. Error bars indicate one standard deviation for a measurement of permittivity. From these comparisons, we calculated weighted correlation coefficients to investigate potential relationships between permittivity and temperature, salinity and brine volume fraction. Weights are the reciprocal of the standard deviation of a measurement. The calculated coefficients are listed in Table 1, with those that are significant at the 5 % level shown in bold.

The physical mechanisms underlying the relationships between low-frequency complex dielectric permittivity; ice properties, including temperature and salinity; and the microstructure of sea ice are more complicated in comparison to single-phase systems such as pure ice or an ionic solution. Studies including those by Scott and Barker (2003), Nordsiek and Weller (2008) and Leroy and Revil (2009), however, present promising results that link the low-frequency behavior of a porous medium to pore volume, connectivity, and geometry. Given the close relationship between ice properties, brine volume fraction and the microstructural characteristics measured, we used principal component analysis (PCA) to identify interrelationships and potential key drivers of variation in measurements of $\varepsilon^*$ (Jolliffe, 2002). A correlation matrix relating all relevant variables including the microstructural characteristics we analyzed is presented in Table 2 with results of the PCA shown in Tables 3, 4, and 5. As only 14 samples are associated with measurements of both microstructural characteristics and complex permittivity, correlations are based on a smaller sample size than the comparison between ice properties and complex permittivity.

## 4. Discussion

### 4.1 Complex permittivity and ice properties

The combined data, as well as that for solely 2013 and 2014, show a consistent significant correlation ($p < 0.05$) between $\varepsilon^{'}$ and ice temperature at frequencies below 1000 Hz. This observation is related to the low-frequency rise in $\varepsilon^{'}$ steepening over the course of the spring season (Figure 5). Less clear is the correlation between $\varepsilon^{'}$ and temperature at 10 kHz for the combined datasets. Establishing causal relationships presents challenges due to temperature having a contrasting effect on the individual electric properties of ice and brine. An increase in temperature in pure ice leads to the weakening of bonds along crystal boundaries and an increase in the concentration of defects resulting in an increase in $\varepsilon^{'}$ for ice (Petrenko and Whitworth, 1999). Conversely, an increase in temperature in an ionic solution such as brine will foster dissociation of cations and anions, increasing hydrate shell shielding of charges and lowering values of $\varepsilon^{'}$.

A significant correlation is found between the imaginary part of the permittivity $\epsilon^{''}$ and temperature at all frequencies. Jones et al. (2012) established that $\varepsilon^{''}$ is closely related to the DC conductivity of the sea ice, which depends on the connectivity of pore spaces. An increase in temperature leads to an increase in the magnitude of $\varepsilon^{''}$ for both ice and brine. For ice, this behavior is related to an increase in the mobility of defects, while for brine, a temperature increase allows ions to respond more readily to the electric field, thus increasing frictional dissipation and resultant dielectric loss.

Although values of $\varepsilon^{'}$ from 2014 gathered below 1000 Hz show a significant negative correlation with bulk ice salinity, there is no significant correlation for either solely 2013 measurements or for all the data combined. Given the greater likelihood of salts being included interstitially between grains, an increase in salinity results in an increase in $\varepsilon^{'}$ of ice (a positive relationship). An inverse relationship exists between salinity and $\varepsilon^{''}$ of brine however, the result of $H_2O$ molecules aligning with additional ions as opposed to the applied electric field. A significant negative correlation exists between $\varepsilon^{''}$ and salinity in 2014 but not in 2013 or for all data combined. This indicates that neither the individual electric properties of ice or brine control the magnitude of $\varepsilon^{''}$ as both are known to display a positive correlation with salinity.

To better understand this finding, the interrelations between ice temperature, bulk ice salinity, and brine volume must be considered. Changes in ice temperature drive the evolution of brine volume fraction and microstructure which, in turn, can influence the evolution of bulk ice salinity. In the winter, low ice temperatures will result in small values of brine volume fraction. Drainage of high salinity brine will be prevented during this time due to low connectivity between pores. In the spring, as ice temperature begins to increase, brine volume fraction will also increase and pores will connect allowing for the drainage of brine and a decrease in bulk ice salinity. As neither temperature nor salinity are found to control measurements of $\varepsilon'$ entirely, brine volume fraction must also be considered. This finding is in agreement with previous studies which have described, primarily qualitatively, the relationship between $\varepsilon'$ and both brine volume fraction and other microstructural characteristics (Addison, 1970; Ingham et al., 2012). As seen in Figure 8 and Table 1, a significant consistent correlation exists between $\varepsilon'$ and brine volume fraction at frequencies below 100 Hz. An increase in brine volume fraction is driven by an increase in temperature, resulting in reduction of both brine and bulk salinity as ice melts and convective overturning and meltwater percolation set in. The positive relationship found between $\varepsilon'$ and brine volume fraction is therefore in agreement with the relationship between $\varepsilon'$ and temperature. Similarly the strong positive correlation at all frequencies between $\varepsilon''$ and brine volume fraction is in agreement with the increase in bulk dc conductivity of sea ice which occurs as brine volume fraction increases – previously observed and modelled by Jones et al. (2010, 2012). How temperature and brine volume fraction may separately influence measurements of $\varepsilon'$ is displayed in Figure 9 where measurements of $\varepsilon'$ at similar temperature but differing brine volume fraction are presented. At temperatures below -5.5 ⁰C, an increase in $\varepsilon'$ is evident as brine volume fraction increases but temperature holds relatively constant. At temperatures above -5.5 ⁰C however, this trend is not as clear given a larger spread in values of $\varepsilon'$. While more data needs to be obtained at temperatures above -5 ⁰C, these findings suggest the impact of brine volume fraction on measurements of $\varepsilon'$ may be different and separate from that of temperature.

**4.2 Relationships between complex permittivity and microstructural characteristics**

Our analysis shows that measurements of apparent permittivity ($\varepsilon^*$) are significantly correlated to brine volume fraction and mean pore volume. Findings from the PCA indicate that the loadings of mean pore volume and bulk brine volume fraction are near-equal for the first principal component responsible for 52% of the variance in measurements of $\varepsilon'$ and 56% of the variance in measurements of $\varepsilon''$. Similar to the above analysis examining the impact of brine volume fraction on $\varepsilon''$, the correlation between $\varepsilon''$ and pore volume (0.836) is expected given the relationship between sea-ice conductivity and pore connectivity (Jones et al., 2012). In processing of tomographic images, brine pores, layers, and channels were not differentiated; therefore, a large "pore" may be composed of a highly connected brine channel. As mean pore volume increases, so too will the connectivity of pore spaces. More pathways will therefore be available for current to flow, increasing conductivity and $\varepsilon''$. This relationship is further supported by a significant correlation (0.486) found between pore volume and the relative depth within each sample at which 25 % of pores are connected to the surface.

The relationship between $\varepsilon'$ and mean pore volume (0.788) is consistent with the positive correlation (0.815) between $\varepsilon'$ and brine volume fraction. The rise in $\varepsilon'$ at low frequencies has been examined in several studies of composite media (e.g., Buchanan et al., 2011; Kemna et al., 2012; Bücker and Hördt, 2013). In sea ice and other porous media, space charge polarization is labelled as the primary cause for this increase below frequencies of about 100 Hz. Dielectric polarization,

responsible for values of $\varepsilon_\infty$ and $\varepsilon_S$ (the high frequency and static values of permittivity) in homogeneous materials, is associated with charge separation at the atomic or molecular scale. Space charge polarization on the other hand, is a broad term associated with the greater separation of mobile charge carriers when an electric field is applied. As a result, the magnitude of $\varepsilon^{'}$ associated with space charge processes is often much greater than that of dielectric polarization (Macdonald, 1953). In the sea-ice system, space charge polarization is challenging to estimate given substantial variations in the proportion of ice to brine with the ice cooling or warming in response to external forcing, and with resultant variations in electric properties (Addison, 1970). In principle, when a low-frequency alternating current is applied, ions within the brine will separate based on charge, creating a concentration gradient that drives diffusion of ions to re-establish balance. The time scale over which diffusive dissipation will occur depends on pore size and shape. Larger pores are therefore associated with longer characteristic relaxation times. This behaviour can result in a peak in $\varepsilon^{'}$ at frequencies below the accepted Debye relaxation of the material.

Connectivity was found to be significantly correlated to both $\varepsilon^{'}$ (0.469) and $\varepsilon^{''}$ (0.654), contributing equally to both first principal components but of lesser magnitude than mean pore volume, brine volume fraction, and temperature. The relationship between $\varepsilon^{''}$ and connectivity is established for laboratory grown and modelled sea ice (Pringle et al., 2009a; Jones et al., 2010). The data we present expand on previous findings and reveal a similar relationship in natural sea ice. The correlation we found between fractional connectivity and $\varepsilon^{'}$ is likely driven by the relationship between connectivity, brine volume fraction, and pore volume. Because temperature has a differing, and opposite, effect on the electric properties of ice and brine, the correlation between temperature and $\varepsilon^*$ is influenced by variations in microstructure that determine the ratio of these two phases.

The absence of a significant correlation between either $\varepsilon^{'}$ or $\varepsilon^{''}$ and the pore surface area to volume ratio (SA/V) is a surprising finding. SA/V was selected as a potential measure of the extent of neck formation within pores. Such features complicate pore geometry and will increase SA/V. Pore necks are a potential source of membrane polarization, in which reduced mobility of a specific type of ion within the fluid in a narrow pore result in a polarization effect (Kemna et al., 2012; Bücker and Hördt, 2013). SA/V ratio has the greatest loading for the second principal component responsible for 20 % of variance in apparent permittivity. Therefore, while a relationship to complex permittivity is not initially evident, measurements of electric properties may still provide a method to track SA/V ratio. Complex conductivity, for instance, is known to be sensitive to the smoothness and distribution of mineral grains (Leroy and Revil, 2009). This relationship is linked to processes occurring on both sides of the electric double layer contributing to values of real and imaginary part of the complex conductivity.

**4.3 Further Analysis**

While the low-frequency dielectric properties of sea ice first became a topic of study in the late 1960s it was not until recently that their possible use as a method to track microstructural evolution has been explored in depth. Studies including that by Buchanan et al. (2011) provide a thorough examination of measurements in the frequency range of 40 Hz to 1 MHz. The laboratory grown ice used in Buchanan et al. (2011)'s study may oversimplify the system however, and does not provide a complete description of sea-ice microstructure. We measure the dielectric permittivity of natural sea ice and its relationships to ice temperature, bulk ice salinity, brine volume fraction, and microstructure. Our results show quantitatively

that at frequencies below about 10-100 Hz the complex dielectric permittivity of sea ice is largely controlled by the evolution of microstructural characteristics. This study provides insight into how permittivity, therefore, can be used as a proxy for the those microstructural characteristics and their evolution.

In hydrogeophysics a significant feature of AC electrical measurements is the existence of a low-frequency polarization which is related to the distribution of grain and/or pore sizes (e.g. Leroy et al., 2008; Revil and Florsch, 2010). Silica grains have a natural negative charge on the surface which results in the establishment of an electrical double layer such that a fixed layer of positive counterions (the Stern layer) is adjacent to the grain surface, with a diffuse layer extending further into the pore fluid. In membrane polarization, as outlined above, it is this electrical double layer that causes a difference in ionic

transport between positive and negative ions when the thickness of the layer approaches the width of a pore. Thus polarization occurs around "necks" in pores with the degree of polarization and its associated relaxation time depending on the distribution of grain/pore sizes. In many cases the relaxation can be observed as a peak in the phase of the complex conductivity (e.g., Joseph et al., 2015) which occurs typically at frequencies between 1 and 0.001 Hz. Determination of the distribution of relaxation times through techniques such as Debye Decomposition (Nordsiek and Weller, 2008) is thus

directly related to the pore size distribution.

Petrenko (1994) has discussed at length the surface conductivity and charge of ice. It is plausible that the existence of a surface charge on ice surfaces adjacent to pores, possibly due to molecular orientation, may also lead to the surface conductivity of sea ice being a function of pore space. If that were the case then low-frequency measurements of

permittivity, or rather conductivity, would in the same way lead to a possible means of direct determination of pore-size distribution. Plotting the phase of the complex conductivity – corresponding to the measured complex permittivity – as a function of frequency suggests that this may indeed be the case. As an example, Figure 10 shows the variation through spring 2013 of the phase of the complex conductivity at depths of 0.15 m and 0.45 m in the ice as function of frequency. At 0.15 m and 0.45 m depth, the rise in $\varepsilon^{'}$ with decreasing frequency shown in Figure 5 manifests at frequencies below

approximately 100 Hz as an increase in the phase of the complex conductivity. The slope of the frequency versus phase curve below 100 Hz and the specific frequency of the phase minimum change as the sea ice evolves in time. Ultimately, as the frequency approaches DC the phase must return to zero; hence, there must also be a phase maximum at a frequency below 10 Hz (the lower limit of our measurements). The change in the low-frequency slope of the phase/frequency curve over the course of the ice season that is apparent in Figure 10 suggests a corresponding change in the frequency and

magnitude of the inferred phase maximum below 10 Hz. At these low frequencies, the phase/frequency relationship may therefore relate to the pore size distribution.

## 5  Conclusions

The physical mechanisms controlling the electric response of sea ice can be inferred through the relationships found between ice properties, microstructure, and the complex dielectric permittivity. We provide quantitative evidence that complex

dielectric permittivity is largely controlled by the evolution of microstructural characteristics. The significant correlation we found between brine volume fraction and the real part of the dielectric permittivity is in agreement with theory that attributes the low-frequency response of sea ice to space charge polarization and interfacial effects. This finding is further supported by the relationship between temperature and salinity and the real part of the permittivity. While the individual correlations

with these two ice properties are statistically significant, the combination indicates that neither ice nor brine dominates the electric response of sea ice. The imaginary part of the dielectric permittivity, known to be related to ionic conductivity, also significantly correlates with brine volume fraction. This finding agrees with previous studies establishing the relationship between DC conductivity and pore connectivity (Jones, 2012)

Development of a proxy relationship capable of accurately inferring key microstructural characteristics of sea ice from complex permittivity measurements is possible. The most relevant characteristic to be targeted first is brine volume fraction. A critical brine volume fraction of 5 % has been identified to mark a percolation threshold above which the bulk of sea ice becomes permeable (Golden et al., 1998). Our results are a first step and suggest that a more detailed examination of changes

in the complex dielectric permittivity across percolation thresholds may improve our broader understanding of evolution of brine volume fraction and key sea ice transport properties. Furthermore, to investigate the applicability of the hydrogeophysical methods mentioned above to monitor brine volume fraction and potentially pore volume, complex permittivity measurements must be extended to frequencies lower than 10 Hz.

Given its use in a number of applications a method to monitor brine volume fraction continuously and in situ will lead to an improved understanding of sea-ice evolution and its impact on the surrounding environment. Measurement of mean pore volume, fractional connectivity, and SA/V ratio may increase the accuracy of these established relationships. Additionally, knowledge of the evolution of specific microstructural characteristics may prove useful in studies beyond those addressing physical properties of the ice. The presence and concentration of biota within the ice, for instance, is largely influenced by

the size and geometry of pore spaces and channels which provide microbiota protection from scavengers while supplying them nutrients (Krembs et al., 2000). The study of oil in sea ice, increasing in its importance, may also benefit from such measurements given their ability to track microstructural evolution on a more detailed level. While approximations can be based primarily on the critical brine volume fraction for upward percolation (Karlsson et al., 2011), improvements need to be made before proxy relationships are established. An improved understanding of microstructural evolution of natural sea ice

including the connectivity of pore space through the ice, the complexity of pathways, and the shape of pores will prove useful as this topic is further examined.

In summary, the research presented here provides a dataset linking the low-frequency dielectric properties of natural sea ice to microstructure and ice properties. Results suggest that measurement of the low-frequency dielectric permittivity offers the

potential to monitor sea-ice microstructure in situ thus improving our current observations of the sea-ice system as a whole. In addition, these findings provide a base of knowledge applicable to a variety of studies and further contribute to our understanding of sea-ice processes on both the macro- and micro-scale.

*Author Contributions:* M. Ingham and H. Eicken are responsible for the project design as well as instrumentation summarized in this paper. M. O'Sadnick and M. Ingham gathered the majority of electric and ice property measurements

while M. O'Sadnick performed the imaging and analysis of microstructural samples. Data analysis and interpretation was a joint effort between M. O'Sadnick, M. Ingham, and H. Eicken with E. Pettit offering valuable feedback and further recommendations. M. O'Sadnick prepared the manuscript with contributions from all co-authors.

*Acknowledgements***:** The authors would like to acknowledge the National Science Foundation for funding support through the CMG Program (OPP-0934683), supplemented by support of the SIZONet Project (OPP-0856867), and the East Asia and Pacific Summer Institutes Program (EAPSI). In addition, this written work was partially supported by the Research Council of Norway PETROMAKS2 program (MOSIDEO- Project Number 243812). Thank you to UIC Science/Umiaq for logistics and field support. We would also like to thank Andy Mahoney, Josh Jones, and Marc Oggier for assistance in collecting mass balance and permittivity measurements.

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

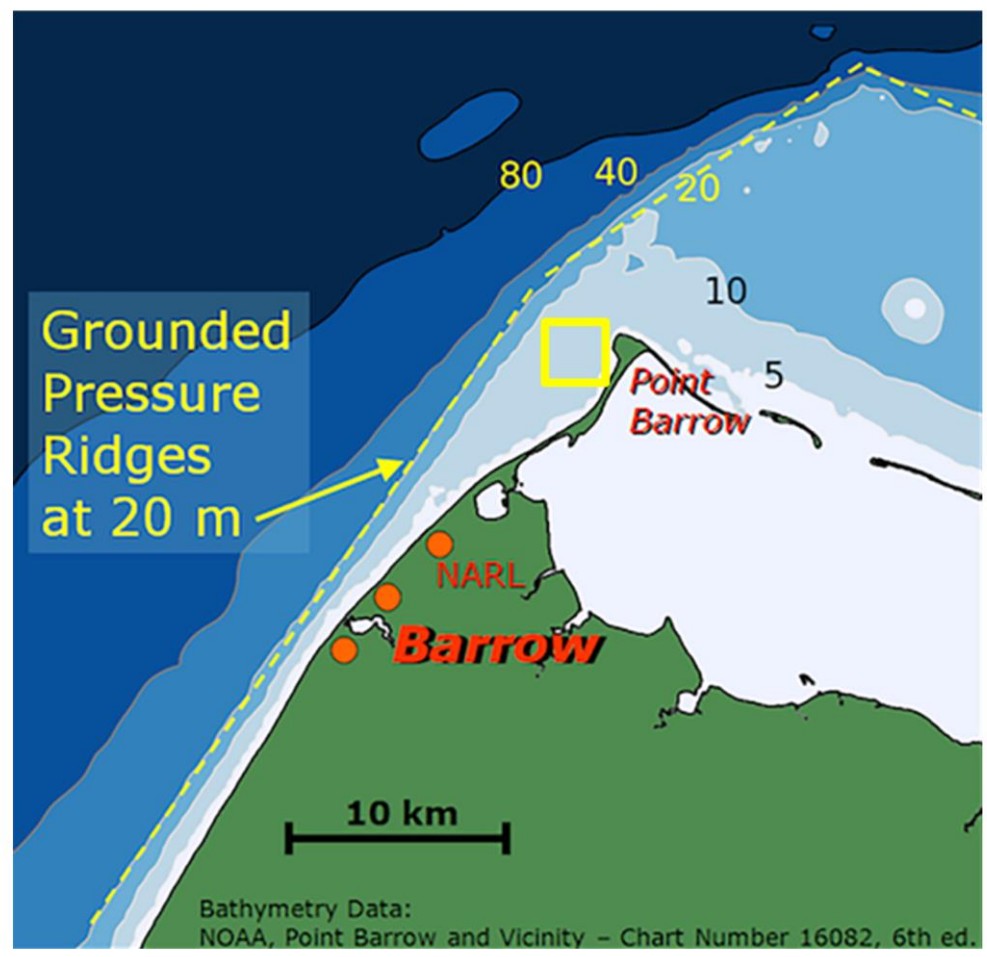

**Figure 1:** Map of Point Barrow area showing the location (yellow box) of the UAF sea-ice mass balance site and permittivity measurements in spring 2013 and 2014. Contours represent bathymetry measured in meters. A grounded pressure ridge is located at approximately 20 m water depth. The water depth at the mass balance site is approximately 7 m.

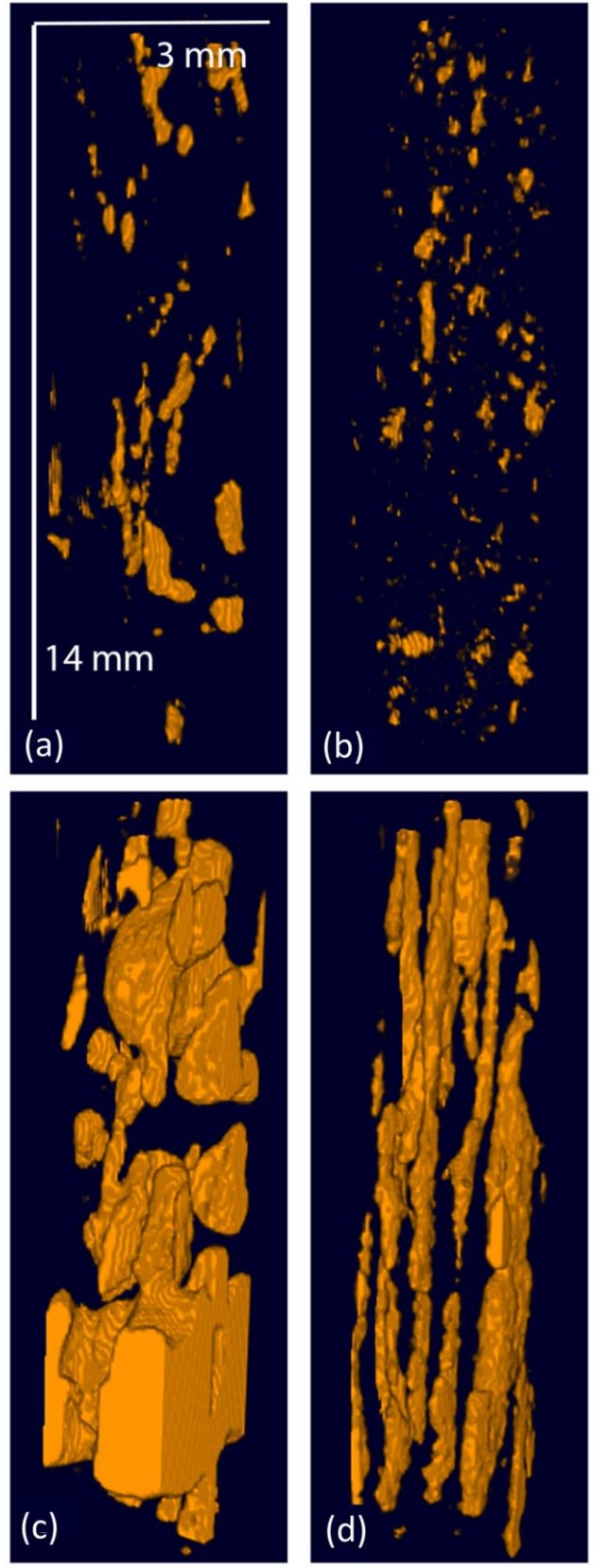

**Figure 2:** 3D images derived using x-ray CT techniques, showing subvolumes of the entire sample. (a) March 2013: 0.20−0.26 m (granular ice), (b) March 2013: 0.88−0.94 m (columnar ice), (c) June 2013: 0.20−0.26 m, (d) June 2013: 0.90−0.96 m.

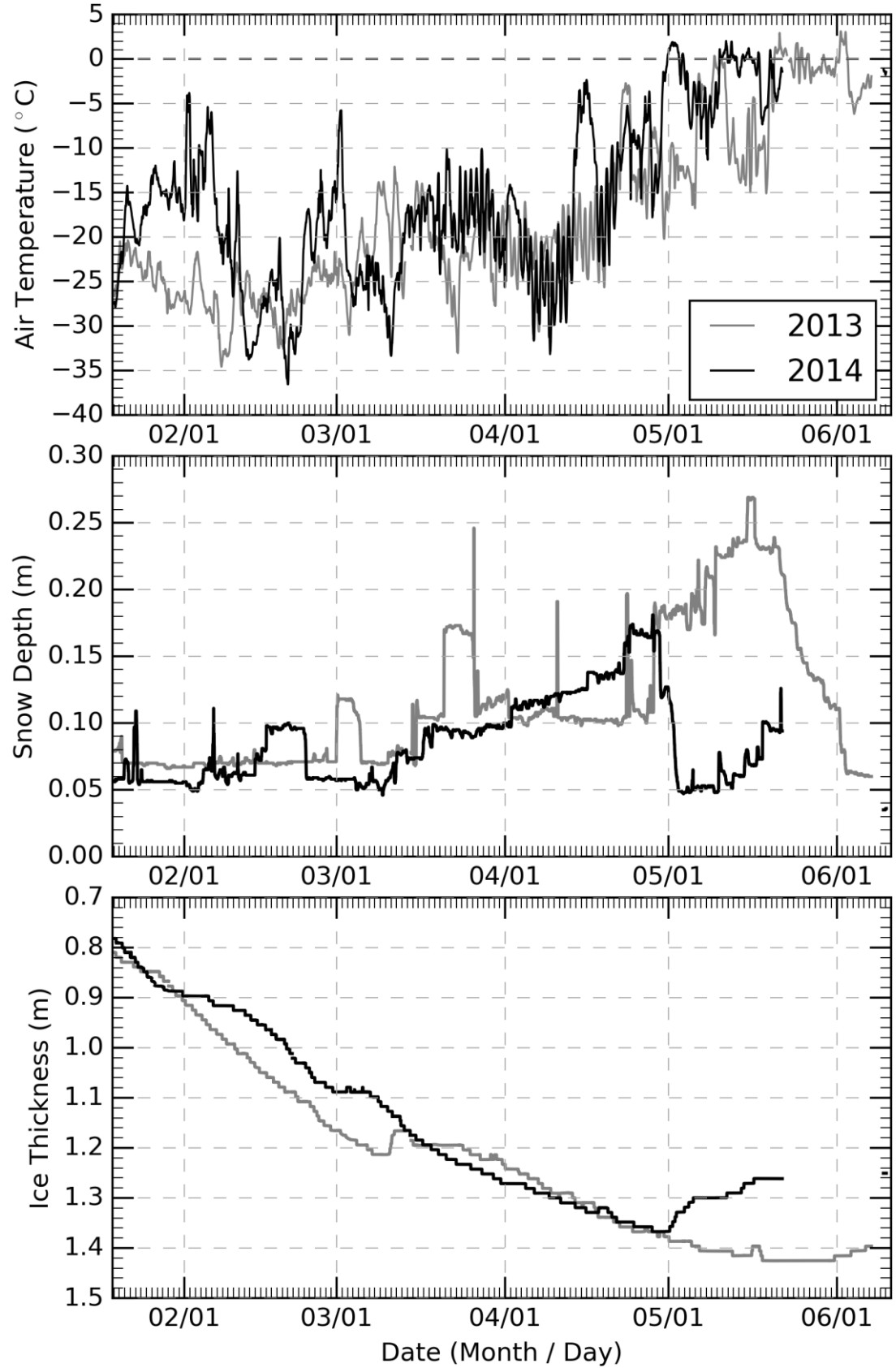

**Figure 3:** Air temperatures, snow depth, and ice thickness for 2013 and 2014 starting 18 January and ending 11 June. 2013 – black solid line, 2014 – gray solid line.

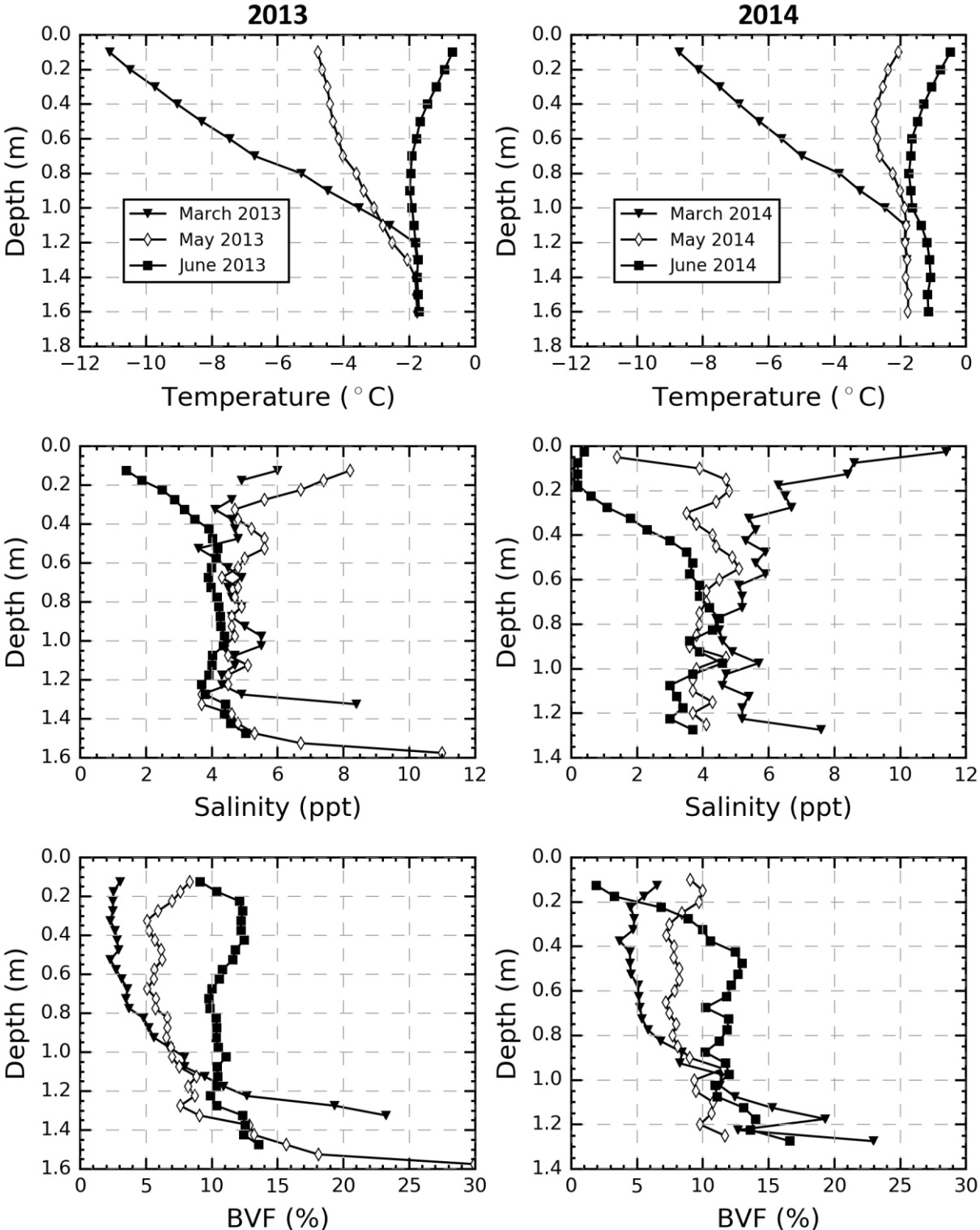

**Figure 4:** Temperature, salinity, and brine volume fraction profiles for spring of 2013 and 2014. March values shown as closed triangles, May values as open diamonds, and June values as closed squares.

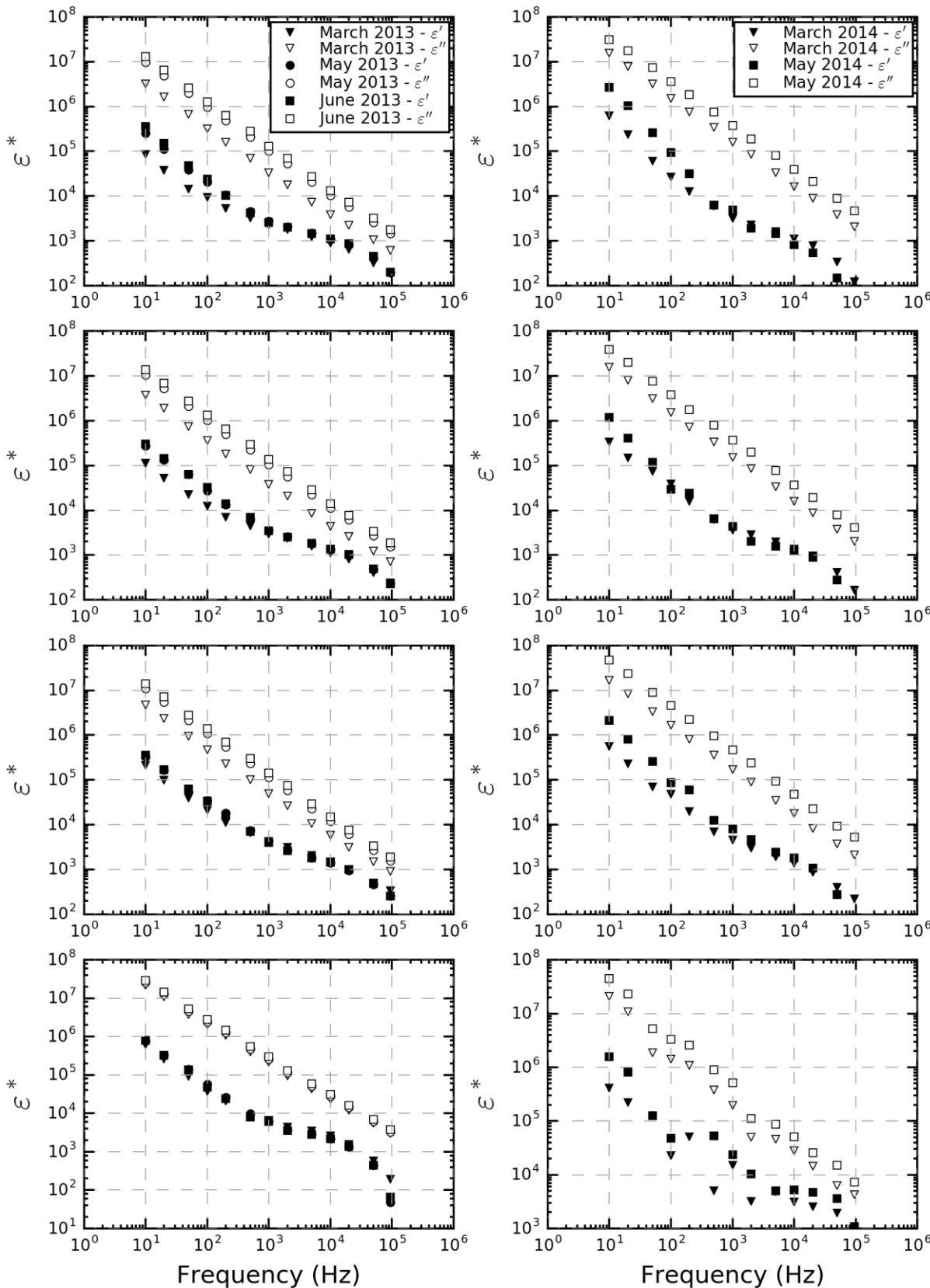

**Figure 5:** Real ($\varepsilon^{'}$, closed symbols), and imaginary ($\varepsilon^{''}$, open symbols) parts of the complex apparent permittivity at different depths. Left hand column shows 2013 data, right hand column 2014 data. Triangles show March data, circles show May, and squares show June data.

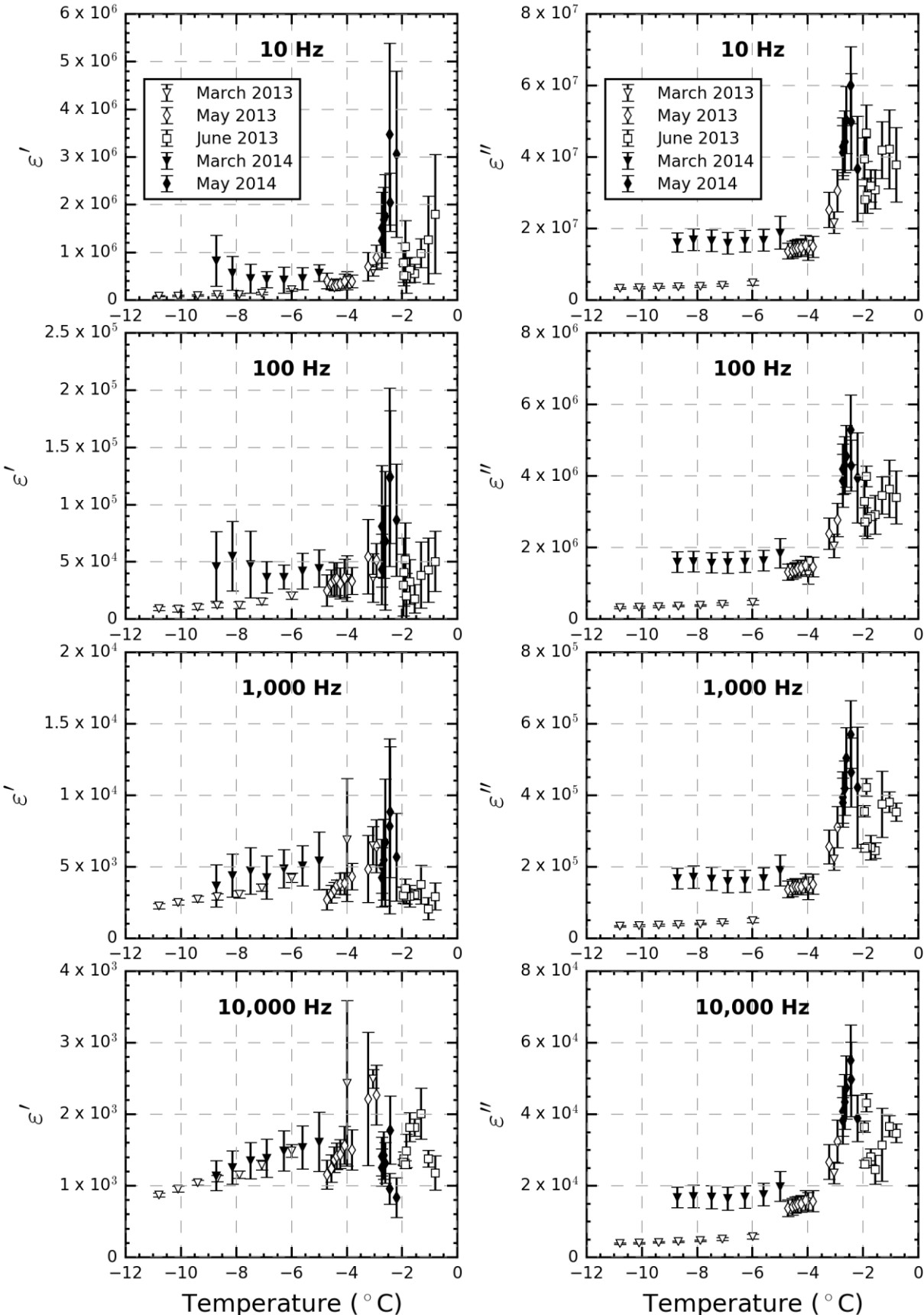

**Figure 6:** Variation of the real ($\varepsilon^{'}$) and imaginary ($\varepsilon^{''}$) parts of the measured permittivity as a function of temperature. Triangles – late March, diamonds – mid May, squares – early June. Open symbols – 2013, filled symbols – 2014.

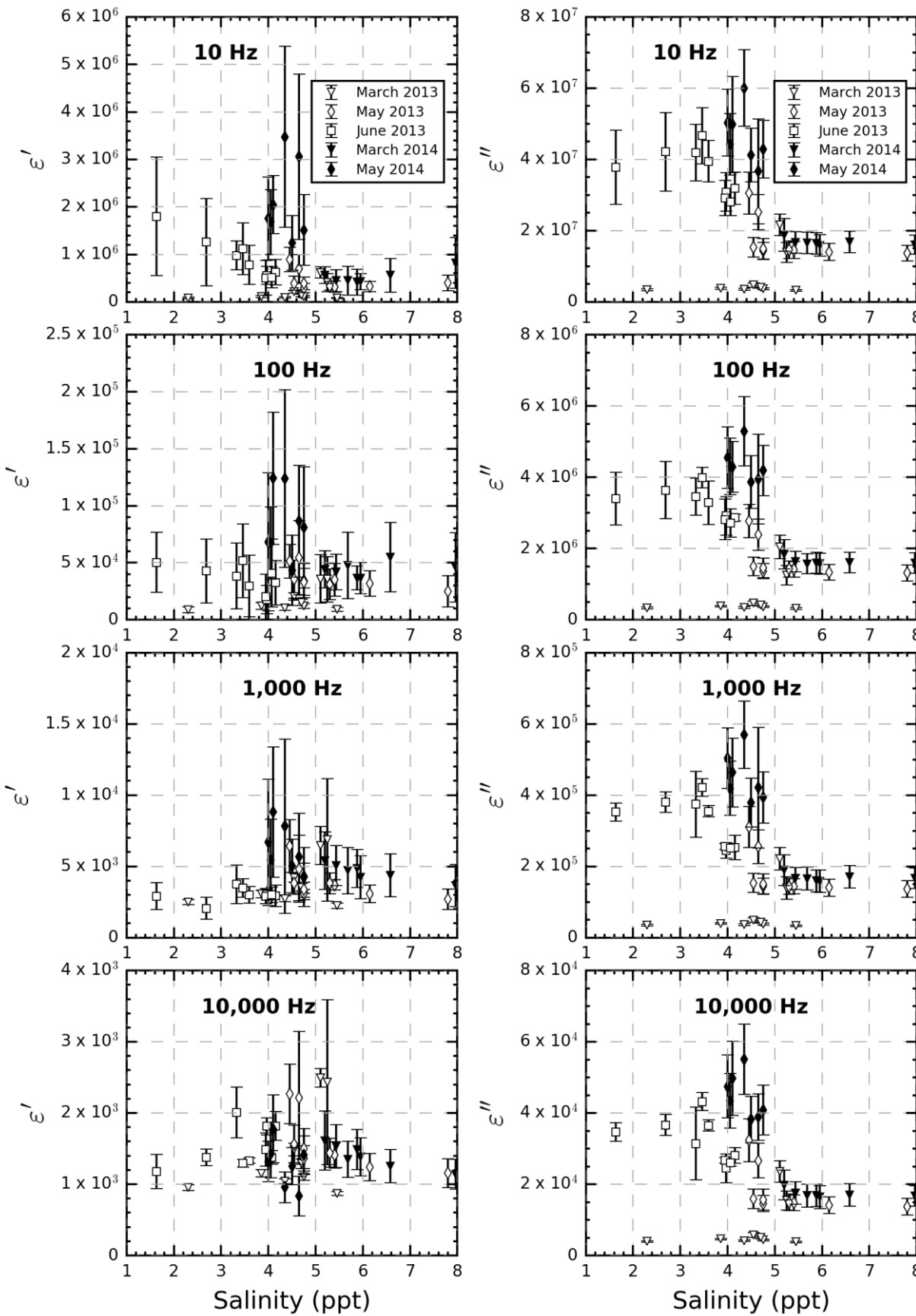

**Figure 7:** Variation of the real ($\varepsilon^{'}$) and imaginary ($\varepsilon^{''}$) parts of the measured permittivity as a function of bulk ice salinity. Triangles – late March, diamonds – mid May, squares – early June. Open symbols – 2013, filled symbols – 2014.

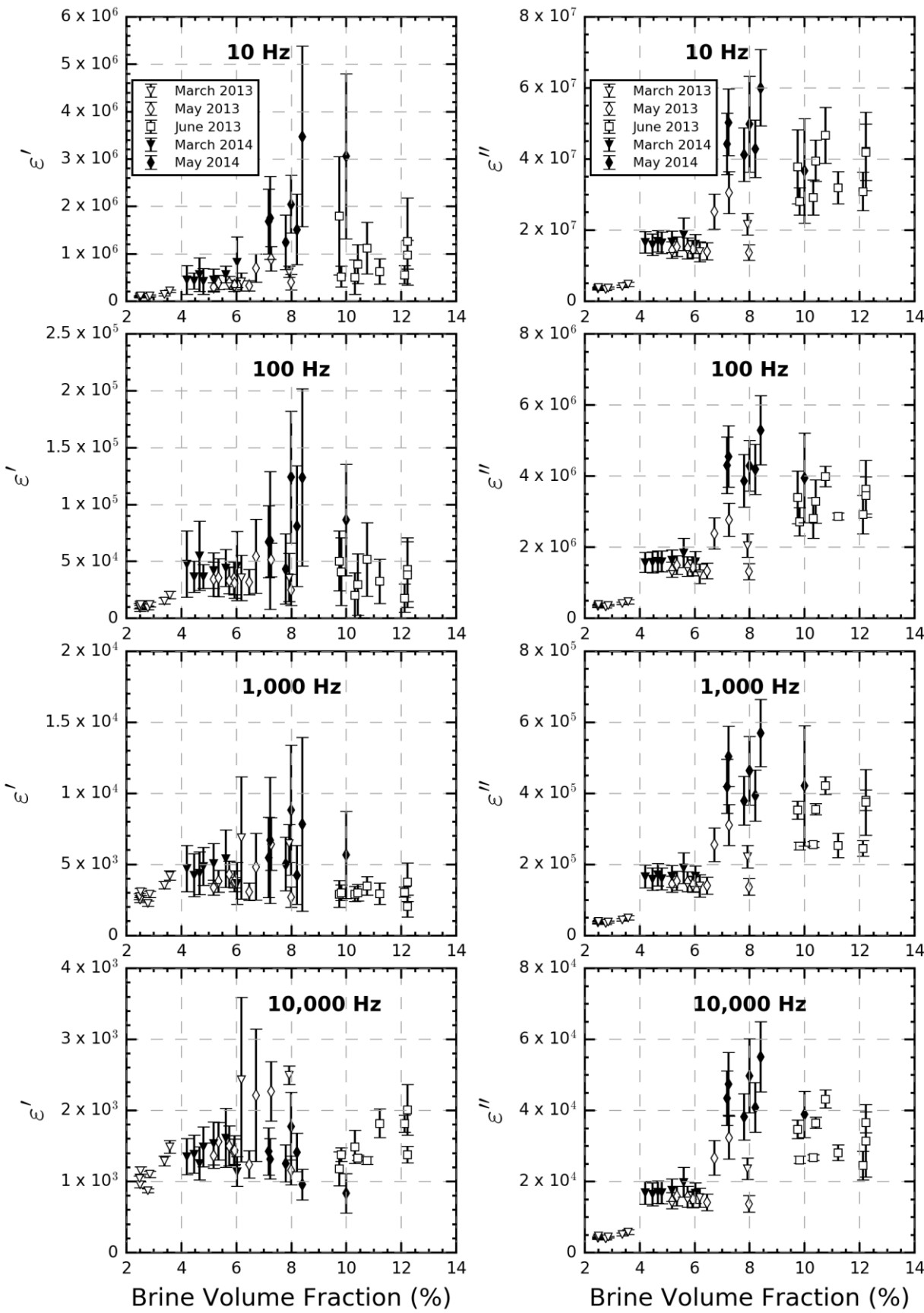

**Figure 8:** Variation of the real ($\varepsilon^{'}$) and imaginary ($\varepsilon^{''}$) parts of the measured permittivity as a function of brine volume fraction. Triangles – late March, diamonds – mid May, squares – early June. Open symbols – 2013, filled symbols – 2014.

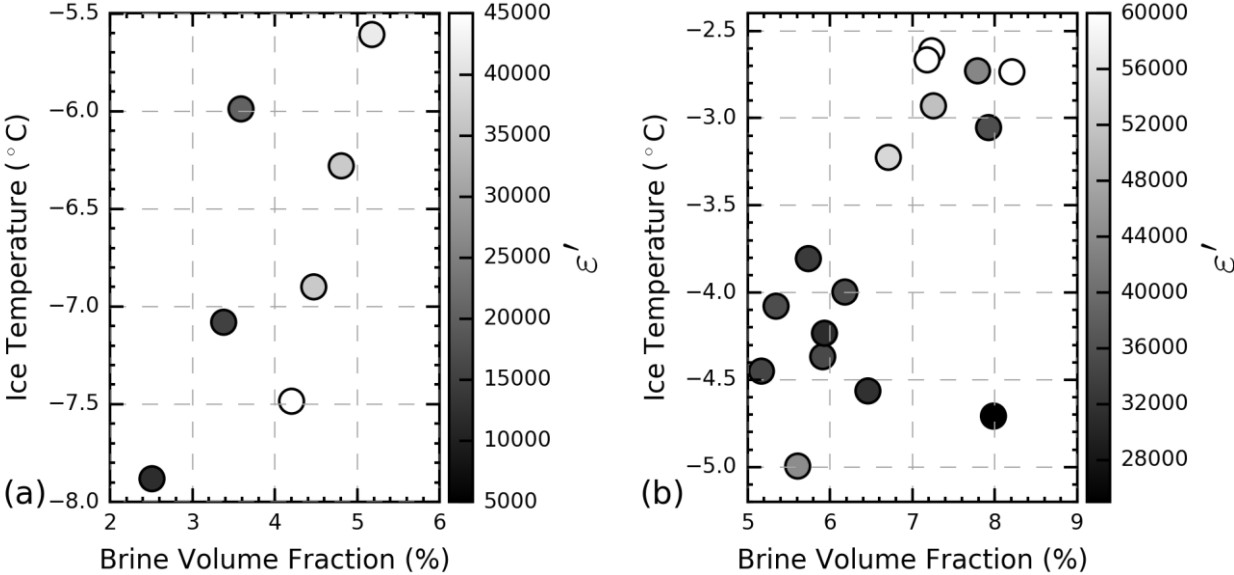

**Figure 9:** The influence of ice temperature and brine volume fraction on measurements of the real part of the complex permittivity ($\varepsilon^{'}$): (a) from 2.0 % to 6.0 % brine volume fraction; (b) from 5.0 % to 9.0 % brine volume fraction.

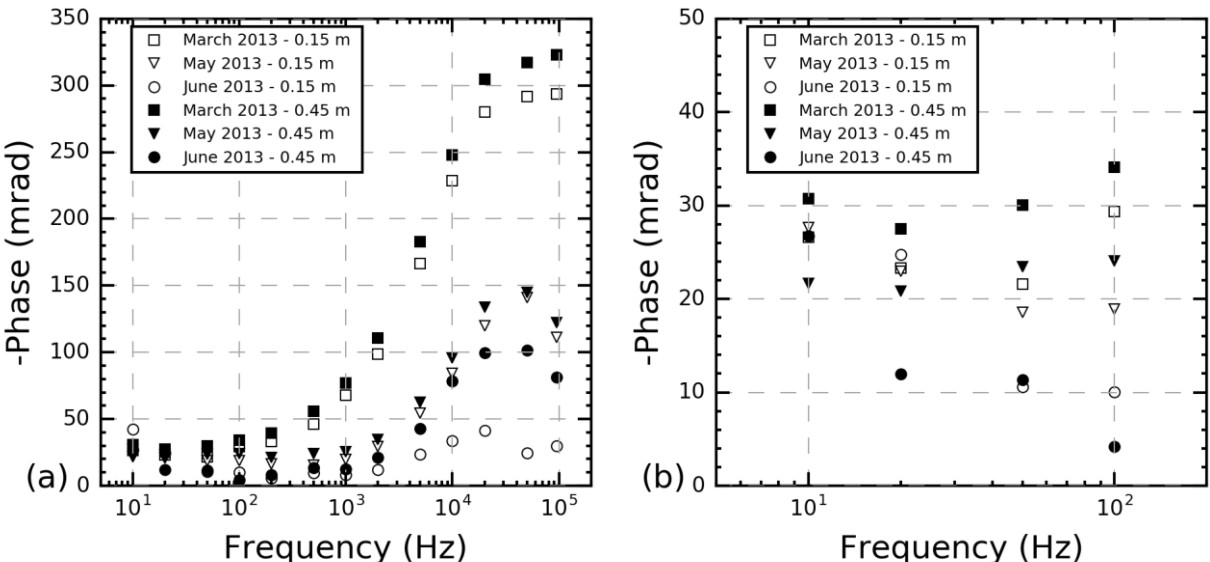

**Figure 10:** Phase of complex conductivity measured in 2013 as a function of frequency: (a) from 0 Hz to 95 kHz; (b) from 0 to 200 Hz..

**Table 1:** Pearson Correlation Coefficients for the real ($\varepsilon'$) and imaginary ($\varepsilon''$) parts of permittivity and ice properties. Coefficients shown in bold are those that are significant at the 5 % level.

| | $\varepsilon'$ - Temperature | | | | $\varepsilon''$ - Temperature | | | |
|---|---|---|---|---|---|---|---|---|
| | 10 Hz | 100 Hz | 1000 Hz | 10000 Hz | 10 Hz | 100 Hz | 1000 Hz | 10000 Hz |
| 2013 | **0.84** | **0.87** | **0.41** | **0.71** | **0.84** | **0.87** | **0.89** | **0.90** |
| 2014 | **0.74** | **0.63** | **0.64** | -0.13 | **0.86** | **0.89** | **0.86** | **0.88** |
| All | **0.70** | **0.71** | **0.39** | **0.66** | **0.81** | **0.84** | **0.86** | **0.87** |

| | $\varepsilon'$ - Salinity | | | | $\varepsilon''$ - Salinity | | | |
|---|---|---|---|---|---|---|---|---|
| | 10 Hz | 100 Hz | 1000 Hz | 10000 Hz | 10 Hz | 100 Hz | 1000 Hz | 10000 Hz |
| 2013 | 0.061 | 0.080 | 0.19 | 0.11 | 0.041 | -0.0050 | -0.13 | -0.12 |
| 2014 | **-0.59** | **-0.53** | **-0.70** | -0.064 | **-0.72** | **-0.74** | **-0.74** | **-0.75** |
| All | 0.094 | 0.17 | 0.24 | 0.11 | 0.12 | 0.072 | -0.041 | -0.030 |

| | $\varepsilon'$ - Brine Volume Fraction | | | | $\varepsilon''$ - Brine Volume Fraction | | | |
|---|---|---|---|---|---|---|---|---|
| | 10 Hz | 100 Hz | 1000 Hz | 10000 Hz | 10 Hz | 100 Hz | 1000 Hz | 10000 Hz |
| 2013 | **0.90** | **0.75** | 0.15 | **0.59** | **0.90** | **0.97** | **0.96** | **0.96** |
| 2014 | **0.90** | **0.74** | **0.44** | -0.49 | **0.85** | **0.89** | **0.88** | **0.86** |
| All | **0.74** | **0.64** | 0.16 | **0.54** | **0.90** | **0.91** | **0.91** | **0.91** |

**Table 2:** Correlation matrix comparing measurements of $\varepsilon'$ and $\varepsilon''$ at 10 Hz to mean pore volume (MVP), surface area to volume ratio (SA/V), depth of 25% fractional connectivity (FC), bulk brine volume fraction (Bulk BVF), sample brine volume fraction (Sample BVF), ice temperature (T), and bulk ice salinity (S).

| | $\varepsilon'$ | $\varepsilon''$ | MPV | SA/V | FC | Bulk BVF | Sample BVF | T | S |
|---|---|---|---|---|---|---|---|---|---|
| $\varepsilon'$ | 1.00 | 0.922 | 0.788 | -0.0580 | 0.469 | 0.82 | 0.786 | 0.67 | -0.24 |
| $\varepsilon''$ | 0.922 | 1.00 | 0.836 | 0.00800 | 0.65 | 0.89 | 0.793 | 0.84 | -0.49 |
| MPV | 0.788 | 0.836 | 1.00 | -0.237 | 0.486 | 0.94 | 0.976 | 0.60 | -0.64 |
| SA/V | -0.058 | 0.00800 | -0.237 | 1.00 | -0.175 | -0.02 | -0.175 | 0.44 | 0.078 |
| FC | 0.469 | 0.65 | 0.486 | -0.175 | 1.00 | 0.45 | 0.430 | 0.53 | -0.47 |
| Bulk BVF | 0.82 | 0.90 | 0.94 | -0.018 | 0.45 | 1.0 | 0.90 | 0.78 | -0.63 |
| Sample BVF | 0.79 | 0.79 | 0.98 | -0.18 | 0.43 | 0.90 | 1.00 | 0.59 | -0.56 |
| T | 0.67 | 0.84 | 0.60 | 0.44 | 0.53 | 0.78 | 0.59 | 1.0 | -0.50 |
| S | -0.24 | -0.49 | -0.64 | 0.078 | -0.47 | -0.63 | -0.56 | -0.50 | 1.0 |

**Table 3**: Results from principal component analysis for $\varepsilon'$ at 10 Hz. Loadings associated with each variable. Mean pore volume (MVP), surface area to volume ratio (SA/V), depth of 25% fractional connectivity (FC), bulk brine volume fraction (Bulk BVF), sample brine volume fraction (Sample BVF), ice temperature (T), and bulk ice salinity (S).

|  | PC1 | PC2 | PC3 | PC4 | PC5 | PC6 | PC7 |
|---|---|---|---|---|---|---|---|
| $\varepsilon'$ | 0.33 | 0.26 | 0.71 | 0.49 | 0.08 | -0.30 | 0.02 |
| MPV | 0.45 | -0.27 | 0.02 | -0.44 | 0.28 | -0.38 | -0.56 |
| SA/V | -0.01 | 0.79 | -0.38 | -0.14 | 0.01 | -0.47 | 0.04 |
| FC | 0.32 | -0.32 | -0.57 | 0.61 | 0.09 | -0.26 | 0.13 |
| Bulk BVF | 0.49 | -0.01 | 0.04 | -0.37 | 0.27 | 0.13 | 0.73 |
| T | 0.44 | 0.38 | -0.17 | 0.16 | 0.12 | 0.69 | -0.36 |
| S | -0.40 | 0.04 | 0.00 | 0.12 | 0.91 | 0.05 | -0.01 |

**Table 4:** Results from principal component analysis for $\varepsilon''$ at 10 Hz. Loadings associated with each variable. Mean pore volume (MVP), surface area to volume ratio (SA/V), depth of 25 % fractional connectivity (FC), bulk brine volume fraction (Bulk BVF), sample brine volume fraction (Sample BVF), ice temperature (T), and bulk ice salinity (S).

|  | PC1 | PC2 | PC3 | PC4 | PC5 | PC6 | PC7 |
|---|---|---|---|---|---|---|---|
| $\varepsilon''$ | 0.42 | 0.16 | 0.04 | -0.01 | 0.79 | -0.43 | 0.02 |
| MPV | 0.43 | -0.26 | -0.35 | 0.32 | -0.32 | -0.32 | -0.57 |
| SA/V | 0.00 | 0.82 | 0.08 | 0.04 | -0.39 | -0.41 | 0.04 |
| FC | 0.31 | -0.30 | 0.84 | 0.08 | -0.25 | -0.17 | 0.13 |
| Bulk BVF | 0.47 | -0.01 | -0.34 | 0.29 | -0.16 | 0.15 | 0.73 |
| T | 0.43 | 0.38 | 0.19 | 0.10 | 0.08 | 0.70 | -0.35 |
| S | -0.38 | 0.04 | 0.14 | 0.89 | 0.20 | 0.02 | -0.01 |

**Table 5:** Results from principal component analysis. Variance associated with each principal component for $\varepsilon'$ and $\varepsilon''$.

| Principal Component | Fraction of variance explained (%), $\varepsilon'$ | Fraction of variance explained (%), $\varepsilon''$ |
|---|---|---|
| 1 | 52 | 57 |
| 2 | 20 | 20 |
| 3 | 11 | 9.3 |
| 4 | 8.7 | 7.2 |
| 5 | 7.2 | 5.9 |
| 6 | 1.4 | 1.2 |
| 7 | 0.30 | 0.30 |