# Peer review of "In situ field measurements of the temporal evolution of lowfrequency sea-ice dielectric properties in relation to temperature, salinity, and microstructure"

_The Cryosphere, 2016_

## Referee Comment (RC1) · Anonymous Referee #1 · 8 Jun 2016

In this manuscript, Megan O'Sadnick and her co-authors present measurements of sea-ice dielectric properties in the frequency range below 100 kHz, which they correlate with independent measurements of sea-ice microstructure.

There are only very few carefully documented, published measurements of low-frequency sea-ice dielectric properties available, which is why this contribution is in principle a welcome addition to the sea-ice literature that can certainly be published subject to some revision.

How much revision is required depends primarily on the intended purpose of this contribution, which is not fully clear to me: if this paper is primarily meant as a paper that describes this particular data set clearly and thus makes it possible for others to make

use of this data, this is a nice contribution that can be published after removal of much of the correlation analysis of section 3

However, if the primary purpose of this contribution is an improved understanding of the relationship between the microstructure of sea ice and its dielectric properties, a major revision is needed that will require further analysis.

Since I do not want to judge the ideal purpose of this paper, I will leave this decision to the authors.

If they were to aim for a paper that provides new understanding, I believe that large parts of section 3 will have to be re-written, for which additional analysis is needed. This is because in my opinion, a focus on correlations is insufficient to provide new understanding on this particular topic. This is because much of the bulk behaviour is well understood (i.e. the impact of T on the dielectric properties of either liquid or solid), and the present analysis remains too superficial to test the robustness of this existing understanding.

For example, sticking to the dependence of epsilon on T, it is already known beforehand that this correlation will be based to a substantial degree on the correlation between the temperature and the brine volume. Hence, a mere correlation with T across all possible measurements of brine volume will be dominated by the changes in brine volume, rather than providing any insights in the role of T for epsilon.

While the authors partly address this issue through their cross-correlation matrix shown in Table 2, this table primarily reflects our existing understanding (high T correlates well with high brine fraction), but does not provide many new insights.

In addition, the extensive work of Buchanan has addressed many of the questions discussed here in more detail, and it remains unclear to me where this work truly goes beyond his existing work.

Hence, in summary, for section 3 it'd be very helpful to have a more concrete overview

of what we know beforehand, whether or not we can test this previous knowledge with the data presented here, and how such tests then provides possibly new insights. For example, sticking to the temperature example, if our current understanding suggests that epsilon' increases with T in pure ice and decreases with T in brine, then it'd be interesting to compare measurements at different T for both the data points with high brine fraction (where epsilon' then should decrease with T if our understanding is correct) and then for the data points with low brine fraction (where epsilon then should increase with T if our understanding is correct). Such more in-depth analysis would then allow us to test more robustly if our current understanding is correct or now.

More detailed comments:

- I really enjoyed reading the intro, background and methods, they were very clear, and extremely well written, I find.

p.4, l.33: How was the gas content of the ice estimated?

p.6, l.8: Only data from March onwards are shown, it seems. It might be good, however, to indeed show data from January to allow one to appreciate the temporal evolution before the first data set.

section 3.2: Error estimates are missing entirely from this section. I doubt, for example, that brine-volume fractions are accurate enough to allow a qualitative statement such as the one given in line 24 on page 6

p.6, l.26-35: (and other places): the reference of comparisons is sometimes not clear. For example, line 30 seems to refer to a spatial increase within the top metre within May (?), while the sentence just before describes a temporal change from March to May. The next sentence then compares the complete uppermost metre in May (?) 2014 to the ice below, rather than the change within the top-most metre. These different comparisons make it sometimes difficult to follow the description.

The same holds, for example, on page 7, l.7-8: First, a temporal comparison between

2013 and 2014 is presented, but the term "similar trends" in the following sentence appears to refer to changes within a single year, which is a bit confusing. It'd be good to check the entire results section for these kind of inconsistencies.

p.6, l.38: I recommend to drop "linear"

p.7, l.14: is this March 2014?

p.7, l.22: I don't think it's necessary to explain the meaning of a significance level

p.7, l.34: "understood" seems too strong, as the following doesn't provide understanding but only the source of the correlation

p.8, l.6: I suggest to start a new paragraph before "A significant"

p.8, l.21: To leading order, the interrelation between T, S and brine volume is not complicated at all, as brine volume is simply given as const*S/T.
* * *

---

## Referee Comment (RC2) · Anonymous Referee #2 · 12 Aug 2016

The authors report in-situ measurements of sea ice in the 10 Hz to 100 kHz range. I really enjoyed reading the important and well-written contribution and admire the methodology, which finally leads to authoritative conclusions. The cross-borehole technique is, compared to the studies of Buchanan et al., suitable to determine the dielectric properties of sea-ice in situ and record their evolution throughout a season. The originality of the paper is to compile these electrical data along with other meteorological, chemical, physical and microstructural properties of ice samples from the nearby vicinity. The named further parameters are determined by means of established standard methods. Therefore, the paper is of high scientific quality. Where the aspect of an appropriate and balanced discussion seems to be fulfilled in relation to the cited literature. To my

knowledge there is no contradicting or ignored work to the field, but my own profile is only in a wider related field. For the same reason I'm more careful when judging the impact, but to my impression the present work can be the impetus for mayor developments to the field.

I agree with referee #1 that if the paper is meant to make a useful dataset available, the correlation analysis is overshooting the mark. On the other hand from reading the title a little bit sloppy I would have expected a discussion like e.g. in Buchanan et al. 2012, where the particular electrical properties, like e.g. extremely high permittivity due to Maxwell-Wagner type relaxations at internal structures, are related to microstructural and chemical properties of the sample. Somehow I had expected to read more about the relation of electrical properties to structural properties by application of suited models in the same way as in Buchanan et al., 2012. But by stating "Implications" the title is conservative and does not promise the "Relations" I had hoped for.From really undertaking this investigation in natural sea ice I expect a lot of pitfalls, as e.g. homogeneity issues. I again agree with referee #1 that for this purpose a lot of more investigation has to be undertaken. On the other hand I find the statistical analysis useful as it is a first step approach to a new field and I enjoyed reading it. In my perception I also agree with referee #1 that the methodology of Buchanan and along the conclusions go much beyond what you have achieved here. But I encourage you to pursue the in situ work in natural sea ice, which is of course far more complicated but when performed in a Buchanan et al. like approach also of far more impacting compared to experiments in lab grown samples. In this sense, your work can be the impetus for much easier estimation of e.g. microstructural properties from electrical measurements. I hope I contributed some more suggestions for your decision how to overwork section 3. And of course encourage you to continue your experiments and pursue their interpretation towards the estimation of microstructural properties of the ice.

---

## Author Comment (AC1) · 11 Sep 2016

Response to Referee #1. Responses are marked by a dash after comments.

In this manuscript, Megan O'Sadnick and her co-authors present measurements of sea-ice dielectric properties in the frequency range below 100 kHz, which they correlate with independent measurements of sea-ice microstructure.

There are only very few carefully documented, published measurements of low frequency sea-ice dielectric properties available, which is why this contribution is in principle a welcome addition to the sea-ice literature that can certainly be published subject to some revision.

- Thank you very much for the thoughtful and thorough comments you offer below. We have taken into consideration your concerns and hope you find our responses and edits are adequate. The manuscript with tracked changes is included as a supplement. Please see below for direct responses to comments.

How much revision is required depends primarily on the intended purpose of this contribution, which is not fully clear to me: if this paper is primarily meant as a paper that describes this particular data set clearly and thus makes it possible for others to make use of this data, this is a nice contribution that can be published after removal of much of the correlation analysis of section 3. However, if the primary purpose of this contribution is an improved understanding of the relationship between the microstructure of sea ice and its dielectric properties, a major revision is needed that will require further analysis. Since I do not want to judge the ideal purpose of this paper, I will leave this decision to the authors.

If they were to aim for a paper that provides new understanding, I believe that large parts of section 3 will have to be re-written, for which additional analysis is needed. This is because in my opinion, a focus on correlations is insufficient to provide new understanding on this particular topic. This is because much of the bulk behaviour is well understood (i.e. the impact of T on the dielectric properties of either liquid or solid), and the present analysis remains too superficial to test the robustness of this existing understanding.

- Our study aimed to take a first step at confirming the ideas and analyses found from laboratory measurements in Buchanan (2011) to field measurements of natural sea ice. While some of the behavior appears well understood in a laboratory environment, we disagree that it is understood fully and confirmed for natural sea ice. Because of the inherent complications of interacting processes in natural sea ice our data show greater spread in impedance and phase measurements at temperatures above -5 degrees C. The correlation analysis presented, therefore, is meant to be a first step in picking apart the different variables and the magnitude of their impact on behavior. We added

further explanation of our motivation for the study but respectfully feel that removing the correlation analysis would change the purpose of the paper- that being to present field measurements of complex permittivity in comparison to T/S/BVF and microstructure and to begin teasing apart the influence of different factors. Such a dataset is the first of its kind and we acknowledge the inevitable shortcomings of field measurements. We have added a more thorough discussion of these issues to further elucidate this point.

For example, sticking to the dependence of epsilon on T, it is already known beforehand that this correlation will be based to a substantial degree on the correlation between the temperature and the brine volume. Hence, a mere correlation with T across all possible measurements of brine volume will be dominated by the changes in brine volume, rather than providing any insights in the role of T for epsilon.

While the authors partly address this issue through their cross-correlation matrix shown in Table 2, this table primarily reflects our existing understanding (high T correlates well with high brine fraction), but does not provide many new insights.

- This is good point, however, we would like to stress that this is, at least partially, the purpose of the principal component analysis. To separate the inherent relationships be-tween T/S/ and BVF and begin to understand the influence of each individual variable. As they are closely intertwined this is not a straightforward process. We acknowledge the need for further study into such topics as the permittivity of ionic solutions of varying salinities and temperatures. However, this itself is not an easy task at low frequencies given the issue of electrode polarization and outside the scope of this work.

In addition, the extensive work of Buchanan has addressed many of the questions discussed here in more detail, and it remains unclear to me where this work truly goes beyond his existing work.

Hence, in summary, for section 3 it'd be very helpful to have a more concrete overview of what we know beforehand, whether or not we can test this previous knowledge with the data presented here, and how such tests then provides possibly new insights. For

example, sticking to the temperature example, if our current understanding suggests that epsilon' increases with T in pure ice and decreases with T in brine, then it'd be interesting to compare measurements at different T for both the data points with high brine fraction (where epsilon' then should decrease with T if our understanding is correct) and then for the data points with low brine fraction (where epsilon then should increase with T if our understanding is correct). Such more in-depth analysis would then allow us to test more robustly if our current understanding is correct or now.

- We have now included a more thorough overview of what was known before hand (particularly at the beginning of section 3.4) and how we have tested and expanded on this knowledge. We agree with your suggestion concerning a direct comparison of such variables as T, BVF, and the real part of the permittivity and have explored options as to the best way to present this analysis. We now will include a plot in the revised manuscript illustrating the correlation between T and BVF and its impact on electric measurements. The plot will show temperature vs. brine volume fraction with markers shaded according to their values for the real part of the permittivity. We plan to explain that though a relationship expected between T and BVF, measurements of permittivity may vary along this curve pointing to the influence of other factors such as microstructural characteristics. The plot will hopefully support and provide a clear visual representation of some of the data presented in the correlation matrix.

More detailed comments:

I really enjoyed reading the intro, background and methods, they were very clear, and extremely well written, I find.

- Thank you!

p.4, l.33: How was the gas content of the ice estimated?

- Gas content was not measured or calculated due to a need for ice density measurements, not taken due to time constraints.

p.6, l.8: Only data from March onwards are shown, it seems. It might be good, however, to indeed show data from January to allow one to appreciate the temporal evolution before the first data set.

- Good catch! This plot will be redone to include data starting in January for both 2013 and 2014 and included in the revised manuscript.

section 3.2: Error estimates are missing entirely from this section. I doubt, for example, that brine-volume fractions are accurate enough to allow a qualitative statement such as the one given in line 24 on page 6

- I believe you may have mistaken a statement of actual brine volume fraction as a percentage of difference. This admittedly was unclear and we have since reworded to provide clarity. In addition, we added to our error estimation provided in section 2.3. Previously we only mentioned temperature and salinity but did not extend to brine volume fraction. This is now included.

p.6, l.26-35: (and other places): the reference of comparisons is sometimes not clear. For example, line 30 seems to refer to a spatial increase within the top metre within May (?), while the sentence just before describes a temporal change from March to May. The next sentence then compares the complete uppermost metre in May (?) 2014 to the ice below, rather than the change within the top-most metre. These different comparisons make it sometimes difficult to follow the description. The same holds, for example, on page 7, l.7-8: First, a temporal comparison between 2013 and 2014 is presented, but the term "similar trends" in the following sentence appears to refer to changes within a single year, which is a bit confusing. It'd be good to check the entire results section for these kind of inconsistencies.

- Thank you, this is a good point to make. We have reworded these sections to enhance clarity.

p.6, l.38: I recommend to drop "linear"

- Done

p.7, l.14: is this March 2014?

- Yes, reworded.

p.7, l.22: I don't think it's necessary to explain the meaning of a significance level

- Taken out

p.7, l.34: "understood" seems too strong, as the following doesn't provide understanding but only the source of the correlation

- Good point, this has been reworded.

p.8, l.6: I suggest to start a new paragraph before "A significant"

- Done

p.8, l.21: To leading order, the interrelation between T, S and brine volume is not complicated at all, as brine volume is simply given as const*S/T.

- Removed word 'complicated' as agree this relationship is itself not so.

Please also note the supplement to this comment:
http://www.the-cryosphere-discuss.net/tc-2016-92/tc-2016-92-AC1-supplement.pdf

---

## Author Comment (AC2) · 11 Sep 2016

- Response to Referee #2. Direct responses are marked by a dash after comment.

The authors report in-situ measurements of sea ice in the 10 Hz to 100 kHz range. I really enjoyed reading the important and well-written contribution and admire the methodology, which finally leads to authoritative conclusions. The cross-borehole technique is, compared to the studies of Buchanan et al., suitable to determine the dielectric properties of sea-ice in situ and record their evolution throughout a season. The originality of the paper is to compile these electrical data along with other meteorological, chemical, physical and microstructural properties of ice samples from the nearby vicinity. The named further parameters are determined by means of established standard methods.

[Figure]

Therefore, the paper is of high scientific quality. Where the aspect of an appropriate and balanced discussion seems to be fulfilled in relation to the cited literature. To my knowledge there is no contradicting or ignored work to the field, but my own profile is only in a wider related field. For the same reason I'm more careful when judging the impact, but to my impression the present work can be the impetus for major developments to the field.

- Thank you for the time you've taken to read through out paper and the thoughtful suggestions!

I agree with referee #1 that if the paper is meant to make a useful dataset available, the correlation analysis is overshooting the mark. On the other hand from reading the title a little bit sloppy I would have expected a discussion like e.g. in Buchanan et al. 2012, where the particular electrical properties, like e.g. extremely high permittivity due to Maxwell-Wagner type relaxations at internal structures, are related to microstructural and chemical properties of the sample. Somehow I had expected to read more about the relation of electrical properties to structural properties by application of suited models in the same way as in Buchanan et al., 2012. But by stating "Implications" the title is conservative and does not promise the "Relations" I had hoped for.

- This is a good point! We have changed the title to provide a more accurate description of the paper's purpose and findings.

From really undertaking this investigation in natural sea ice I expect a lot of pitfalls, as e.g. homogeneity issues. I again agree with referee #1 that for this purpose a lot of more investigation has to be undertaken. On the other hand I find the statistical analysis useful as it is a first step approach to a new field and I enjoyed reading it. In my perception I also agree with referee #1 that the methodology of Buchanan and along the conclusions go much beyond what you have achieved here. But I encourage you to pursue the in situ work in natural sea ice, which is of course far more complicated but when performed in a Buchanan et al. like approach also of far more impacting
compared to experiments in lab grown samples. In this sense, your work can be the impetus for much easier estimation of e.g. microstructural properties from electrical measurements. I hope I contributed some more suggestions for your decision how to overwork section 3. And of course encourage you to continue your experiments and pursue their interpretation towards the estimation of microstructural properties of the ice.

- In response to your and the other reviewer's comments we have now made changes to the manuscript to further explain the motivation of our work and correlation analysis as well as how we have advanced on Buchanan's findings. The manuscript with tracked changes is included as a supplement. I hope you find our edits address your concerns adequately.

Please also note the supplement to this comment:
http://www.the-cryosphere-discuss.net/tc-2016-92/tc-2016-92-AC2-supplement.pdf

**Supplement:**

**In situ  measurement of low-frequency sea-ice dielectric properties in relation to ice properties and  microstructure**

Megan O'Sadnick[1,4], Malcolm Ingham[2], Hajo Eicken[3], Erin Pettit[1]

[revised manuscript text omitted]

Here we explore the relationship between low-frequency complex dielectric properties of sea ice with ice properties such as temperature, salinity and brine volume fraction and specific aspects of ice microstructure. We present the first in situ measurements of the seasonal variation of low-frequency complex permittivity of natural sea ice; measured at Barrow, Alaska in 2013 and 2014. Although Ingham et al. (2012) presented measurements of the complex permittivity of Antarctic sea ice, they did not track the temporal evolution. We also show; associated measurements of ice properties including temperature, salinity and brine volume fraction; and microstructural characterization of sea-ice samples gathered in parallel with the impedance measurements. After noting the manner in which the frequency dependence of complex  permittivity varies seasonally,  to improve our understanding of the physical mechanisms dominating the frequency dependence and seasonal evolution, we, then, 
[revised manuscript text omitted]

|---|---|---|
| 1 | 52 | 57 |
| 2 | 20 | 20 |
| 3 | 11 | 9.3 |
| 4 | 8.7 | 7.2 |
| 5 | 7.2 | 5.9 |
| 6 | 1.4 | 1.2 |
| 7 | 0.30 | 0.30 |

---

## Author Response (AR2)

**Author's Response to Editor's Decision : Publish subject to technical corrections**

Dear Dr. Eisen,

Thank you for your feedback on our manuscript. Below we address your comments.

- "Ice extent ranges between 5 ...": This number is outdated, especially the minimum is an increasingly smaller number. Please adapt and use recent review on sea ice for citation.
  - This has now been changed to reflect the latest estimates- "3.4 and 15 million km2 in the Arctic and 2.3 and 20 million km2 in the Antarctic (Fetterer et al., 2016) "
- Figures: most of them would benefit from major and minor tick marks. For Figure 3 this is mandatory. Include start and end date in caption of Fig. 3
  - All figures have been updated to include major and minor tick marks as well as a grid to better guide the reader
- Figure 1: explain numbers on water depth in caption for clarification; e.g. "Ridge at 20 m": keel depth, water depth, ridge height?
  - Caption changed to better describe figure- "Map of Point Barrow area showing the location (yellow box) of the UAF sea-ice mass balance site and permittivity measurements in spring 2013 and 2014." Changed to: "Map of Point Barrow area showing the location (yellow box) of the UAF sea-ice mass balance site and permittivity measurements in spring 2013 and 2014. Contours represent bathymetry measured in meters. A grounded pressure ridge is located at approximately 20 m water depth. The water depth at the mass balance site is approximately 7 m."
- Figure 6: something is wrong with the unit °C, at least in the pdf.
  - Degree symbol was not superscripted. Corrected in all figures where temperature is mentioned.

[revised manuscript text omitted]